

# Anthropogenic influences on the distribution of a threatened apex-predator around sustainable-use reserves following hydropower dam installation

Deborah C. Dávila Raffo[1,2], Darren Norris[2,3,4], Sandra Maria Hartz[1] and Fernanda Michalski[2,4,5]

[1] Postgraduate Programme in Ecology, Federal University of Rio Grande do Sul, Porto Alegre, Rio Grande do Sul, Brazil
[2] Ecology and Conservation of Amazonian Vertebrates Research Group, Federal University of Amapá, Macapá, Amapá, Brazil
[3] School of Environmental Sciences, Federal University of Amapá, Macapá, Amapá, Brazil
[4] Postgraduate Programme in Tropical Biodiversity, Federal University of Amapá, Macapá, Amapá, Brazil
[5] Pro-Carnivores Institute, Atibaia, São Paulo, Brazil

Corresponding author
Fernanda Michalski,
fmichalski@unifap.br

## ABSTRACT

Although previous studies demonstrate declines in many large bodied species following hydropower dam installation, others suggest that some species, including the giant otter (*Pteronura brasiliensis*) may return to newly created reservoir habitats. Yet, there is a lack of evidence to support such theories. Here, we analyzed the effects of a hydropower dam on giant otters using a before-after control-impact study design in the eastern Brazilian Amazon. We collected data 3 years before (2011–2012 and 2015) and after (2017–2019) dam construction, totaling 6,459 km of surveys along rivers with 43 direct sightings of giant otters. Contrary to expectations, our results revealed that giant otters did not remain nor did they return to the dam reservoir. Beyond the zone directly impacted by the dam giant otter occurrence and relative abundance declined next to sustainable-use reserves where the river was more intensely used by people. Our study showed that in the absence of active management sustainable-use reserves and low human density were not sufficient to maintain this apex-predator. Our findings suggest a need to proactively create and maintain areas with low levels of anthropogenic disturbance to enable sustainable coexistence between energy demands and biodiversity across Amazonia.

## INTRODUCTION

The range and populations of large bodied freshwater species are declining rapidly due to anthropogenic changes impacting freshwater environments (*He et al., 2019*; *Tickner et al., 2020*). Energy demand has generated an unprecedented expansion of new and planned hydropower dams across Amazonia (*Anderson et al., 2019*; *Finer & Jenkins, 2012*). Hydropower developments are among the most significant threats to Amazonian

freshwater ecosystems (*Castello & Macedo, 2016*; *Latrubesse et al., 2017*) and biodiversity (*Dudgeon, 2019*; *Latrubesse et al., 2021*; *Vasconcelos et al., 2021*). Thus, even the large coverage of protected areas (22%), which together with indigenous territories protects more than half (52%) of the Amazon rainforest (*Walker et al., 2020*), is unlikely to offset the negative effects caused by hydropower dams. Although adverse dam effects on Amazonian vertebrates have been widely suggested, there remains a lack of robust evidence to inform development of effective minimization and mitigation actions (*Santos, Michalski & Norris, 2021*). To our knowledge, here we provide the first study on the effects of a new hydropower dam on a threatened apex-predator (giant otter—*Pteronura brasiliensis*) using a before-after control-impact (BACI) study design (*Conner et al., 2016*).

Most work documenting biological responses to hydropower impacts focused on fishes in temperate regions (*Algera et al., 2020*; *Harper et al., 2020*). Although some localized studies quantified hydropower dam impacts on vertebrates such as freshwater turtles (*Bárcenas-García et al., 2022a*) and dwarf caimans (*Paleosuchus palpebrosus* and *Paleosuchus trigonatus*) in the Brazilian Amazon, the majority focused on fishes, and adopted less robust study designs that limited the insight possible from the available scientific evidence (*Santos, Michalski & Norris, 2021*). It is therefore unsurprising that to date, there has been no consensus on the effects of hydropower dams on giant otters, the world's largest otter (*Groenendijk et al., 2021*; *Santos, Michalski & Norris, 2021*).

There are currently 29 hydropower dams (with installed power > 30 MW) across 5 Mkm$^2$ of the Legal Brazilian Amazon, yet a recent review found studies of giant otter at only two dams (*Santos, Michalski & Norris, 2021*). Although there are few studies around dams it is possible to suggest likely impacts based on previous studies. Giant otters are large, piscivorous group living otters that were once widely distributed across South America, but since 2,000 are classified as Endangered by the IUCN Red List Criteria (*Groenendijk et al., 2021*). Giant otters are sensitive to a variety of anthropogenic perturbations including: deforestation surrounding waterways (*Duplaix, 2002*; *Michalski et al., 2006*; *Rosas, Mattos & Cabral, 2007*), traffic of water-craft (*Duplaix, 2002*; *Staib & Schenck, 1994*), river pollution, and human exploitation of fishes (*Carter & Rosas, 1997*; *Duplaix, 2002*). The species is a visual predator and previous studies suggest that water transparency is an important factor that can affect both habitat use and distribution of giant otters (*Duplaix, 1980*; *Duplaix, Evangelista & Rosas, 2015*; *Rosas, Zuanon & Carter, 1999*). Giant otters also use terrestrial habitats, including areas close to river and lake margins to build dens and maintain campsites that are important habitats for this group living otter (*Carter & Rosas, 1997*; *Duplaix, 1980*). While water transparency can be higher in the dam (*de Castro et al., 2021*) the vulnerability to permanently flooding in dams (*Norris, Michalski & Gibbs, 2018*) and the amplitude of the water level fluctuations can affect the availability of dens and campsites for giant otters (*Duplaix, Evangelista & Rosas, 2015*). Studies argued that giant otters can inhabit hydropower reservoirs (*Calaça, Faedo & de Melo, 2015*; *Rosas, Mattos & Cabral, 2007*), and could even benefit with the temporal increase of fish abundance in the reservoir (*Calaça & de Melo, 2017*). In contrast, others suggest that the reservoir environment provides relatively low-quality aquatic habitats and limited food web to sustain these apex predators (*Palmeirim, Peres & Rosas,*

*2014*). A more recent notice found no evidence that giant otters remain in the region directly impacted by another dam (*Michalski & Norris, 2021*).

Here, we provide the first evidence using a robust study design that even in the most preserved state of the Brazilian Amazon (*INPE, 2021*), sustainable-use reserves are not effective to reduce the pervasive effects of hydropower dams. If dam reservoirs that held giant otters before dam construction could have potential for the conservation of the species (*Rosas, Mattos & Cabral, 2007*), we expected sightings of the species in the reservoir during 3 years of monitoring following dam construction. Contrary to expectations we confirmed that giant otters did not remain nor return to a new reservoir 3 years after construction. Additionally, changes following the new hydropower dam, including movement of riverside populations to previously uninhabited areas negatively affected the distribution of giant otters. This large bodied apex-predator was found to depend on more isolated areas, both further from hydropower dams and with a low human density.

## MATERIALS AND METHODS

### Ethics statement

Data collection used non-invasive, field observation and did not involve direct contact or interaction with animals. Fieldwork was conducted under research permit number IBAMA/SISBIO 26653, 49632, and 69342 to DN and FM, issued by the Instituto Chico Mendes de Conservação da Biodiversidade (ICMBio).

### Study area and data collection

The data were collected along 212.7 km of rivers that flow between two sustainable-use protected areas (Amapá National Forest and Amapá State Forest) in the eastern Brazilian Amazon (Fig. 1) (*Michalski et al., 2020*). According to the International Union for Conservation of Nature (IUCN) criteria these two protected areas are classified into multiple use management category VI (*Dudley, 2008*). Brazilian sustainable-use reserves include areas with different management objectives that contain traditional human communities and need to ensure both, the rights of local communities to explore natural resources and the maintenance of biodiversity and ecological processes (*MMA, 2000*; *Peres, 2011*). These protected areas and the study rivers are upstream from three dams installed along an 18 km stretch of the Araguari River (*SIGEL, 2021*). The present study focuses on the 219 MW Cachoeira Caldeirão dam (51°17′W, 0°51′N), which became operational in 2016 and is the most recent and most upstream dam in the Araguari River. *Oliveira, Norris & Michalski (2015)* demonstrated the presence of giant otters in upstream rivers prior to the filling of the Cachoeira Caldeirão dam reservoir. The reservoir was filled in 2016 and covers an area of 47.99 km$^2$, including 24 km$^2$ of flooding beyond the original river channels. We combined collections that occurred during the low (September-November) and rising (December-February) river levels (*Oliveira, Norris & Michalski, 2015*) in two periods, before (2011–2012, 2015) and after (2017–2019) dam construction. On a coarse scale, the river stretches were divided into three zones, one directly impacted by the dam reservoir and two control zones upstream of the direct dam impacts (Fig. 1). The 59.1 km dam "reservoir" impact zone included approximately 39 km
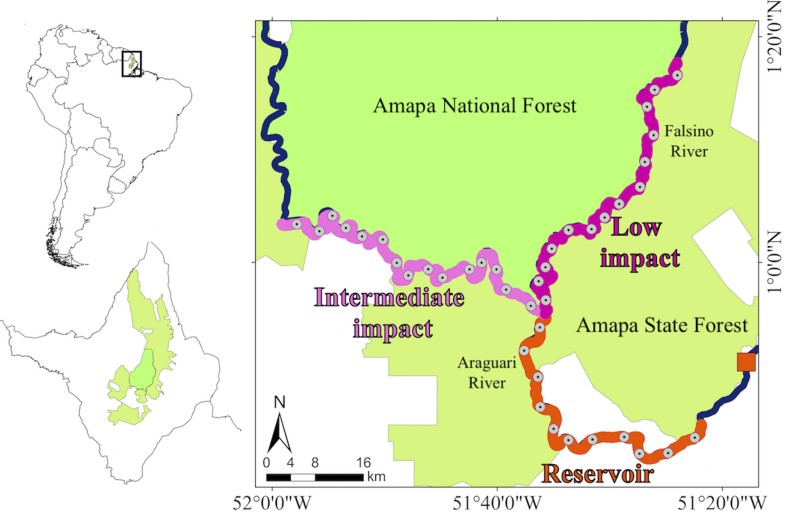

**Figure 1 Study area and giant otter distribution.** Study area with insets showing location of Amapá state and protected area coverage within the state. The Polygon of the Amapa National Forest and the Amapa State Forest showing the zones that were sampled, the location of the Cachoeira Caldeirão hydropower dam (orange square), the division of the Araguari and Falsino rivers into three zones (reservoir, intermediate impact and low impact) and the central point of each subzone.

of reservoir and an additional 20 km where upstream river levels changed after dam construction (*Bárcenas-García et al., 2022a*; *Norris, Michalski & Gibbs, 2018*), together with relatively high levels of anthropogenic disturbance. The control zones included a 79.3 km "intermediate impact" zone not directly affected by the dam (>70 km upstream of the dam) but with an intermediate anthropogenic disturbance (including commercial fishing activity), and a 74.3 km "low impact" zone not directly affected by the dam (>70 km upstream of the dam) with low anthropogenic disturbance and no commercial fishing allowed (*Oliveira, Norris & Michalski, 2015*). On a finer scale, each zone was divided into subzones of similar river-length (mean ± SD = 5.31 ± 0.052 km, min-max = 5.24 – 5.40 km). The subzone river length corresponds approximately to the minimum home range estimated in the dry season for giant otters in the Brazilian Amazon (4.6 km) (*Evangelista & Rosas, 2011*).

Diurnal census surveys conducted in a 9 m aluminum boat at a standard speed (mean ± SD = 9.6 ± 2.8 km/h, min-max = 1.4 – 15.9 km/h), by a team of a minimum of two people, were used to search for direct sightings of giant otters along the river stretches (*Oliveira, Norris & Michalski, 2015*). To minimize any possible observer related bias between years, a local resident with over 20 years of knowledge on otters, was present in our searches throughout the entire study period. The principal objective was to monitor temporal and spatial patterns not social behavior. As such, observations of giant otters were limited to a 15 min maximum that prevented observation of additional details such as feeding, social behavior, and group size that require extended acclimatization of the study species. Locations of riverside houses and number of boats during the study period were also obtained during our boat census surveys with the aid of a handheld GPS.

As preliminary analysis showed that the number of boats was highly correlated with the number of houses (*Oliveira, Norris & Michalski, 2015*) and previous studies showed that riverside people in the region usually stay close (<500 m) to their homes when fishing (*Norris & Michalski, 2013*) we excluded 'boats' in further analysis.

**Data analysis**

Two response variables were analyzed: (1) presence, and; (2) relative abundance (frequency of sightings by the sampling effort (km)) of giant otters (electronic Supplemental Material, Raw Data). To determine the distance along the river between the central point of each subzone, and the nearest house we used functions available in the riverdist package (*Tyers, 2017*). Spatial autocorrelation was quantified using Mantel tests (*Dale & Fortin, 2014*), which confirmed that each subzone could be considered as spatially independent sample units ($p = 0.995$ and $p = 0.797$, for giant otter presence and relative abundance, respectively).

Generalized additive models (GAMs) were applied to all model sets using the mgcv package (*Wood, 2017*) in R (*R Core Team, 2020*). To represent the before-after control-impact sampling design (*Underwood, 1993*), the response variables were modeled by including two parametric fixed factors (before-after, zone and their interaction). A null model (intercept only), effort model (sampling effort) and a full model (with all explanatory variables) were also included for comparisons. GAMs were run with binary and tweedie error distribution family for giant otter presence and relative abundance, respectively. All models included subzone location as a random effect to account for unexplained variation that was not part of the sampling design and enable estimation of the mean and variance of the global distribution of explanatory variables (*Pedersen et al., 2019*). Model comparison was performed using a second-order Akaike information criteria corrected for small sample sizes (AICc), with an evidence ratio ≥ 2 ($\Delta$AICc ≤ 2) used to identify the most plausible models (*Burnham & Anderson, 2002*). Data used for all models are available in the Supplemental Material.

## RESULTS

During 6 years and 6,459 km of survey effort there were 43 giant otter sightings. Giant otters were recorded in less than half of the subzones (40%, 16/40) and were rarely encountered in the reservoir zone even before dam construction (Fig. 2). Prior to dam construction the low impact zone had the majority of giant otter detections (59%, 16/27) and there was a threefold increase in relative abundance in the low impact zone compared with the reservoir zone before dam construction (Table 1).

After dam construction giant otter relative abundance increased slightly in the least impacted zone (Fig. 2), but changes did not follow the same patterns across the three zones. Indeed, after dam construction there was a significant difference in the proportion of subzones with giant otters between zones ($\chi^2 = 17.9$, df = 2, $p = 0.0001$), with declines in giant otter presence and relative abundance in the reservoir and intermediate impact zones (Fig. 3). In the low impact zone there was no significant change in presence but differences in relative abundances did vary between subzones, with proportionally more
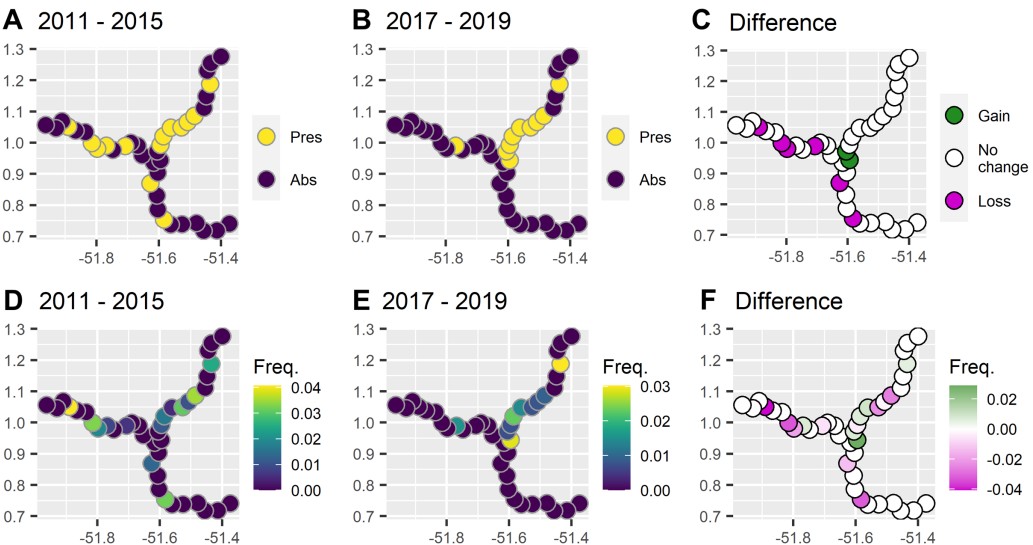

**Figure 2** **Comparison between presence and frequency of giant otter sightings before and after the dam became operational.** (A) Presence of giant otter before and (B) after the dam became operational, and (C) subzones where giant otter sightings disappeared and where new sightings were registered. (D) Frequency of sightings before and (E) after the dam was operational, and (F) subzones where the frequency of sightings decreased or increased.

**Table 1** **Summary of survey effort. Sampling effort and number of sightings of giant otters for the three sampling zones between January 2011 and December 2019.**

|  | Zone | | |
|---|---|---|---|
|  | **Reservoir** | **Intermediate impact** | **Low impact** |
| River km | 59.1 | 79.3 | 74.3 |
| Subzone count | 11 | 15 | 14 |
| Census km (before/after) | 1,523 (822/701) | 2,214 (1,396/819) | 2,722 (1,459/1,263) |
| House (before/after) | 105/89 | 11/28 | 3/7 |
| Giant otter (before/after) | 3 (3/0) | 9 (8/1) | 31 (16/15) |
| Giant otter relative abundance (before/after) | 0.002 (0.004/0) | 0.004 (0.006/0.001) | 0.011 (0.011/0.011) |

subzones with increases in the low impact zone compared with the other two zones ($\chi^2 = 7.5$, df = 2, $p = 0.024$, Fig. 3).

The model comparison analysis showed that the changes in both responses were best explained with the inclusion of all variables. Models with only a subset of variables in isolation (*e.g.*, effort, zone) were not as strongly supported based on the differences in model deviance explained and AICc values (Table 2).

# DISCUSSION

The analysis of this long-term study with a unique standardized before-after control-impact dataset revealed that giant otters do not remain nor return to a reservoir with anthropogenic disturbance. Previous studies that demonstrated the reestablishment of

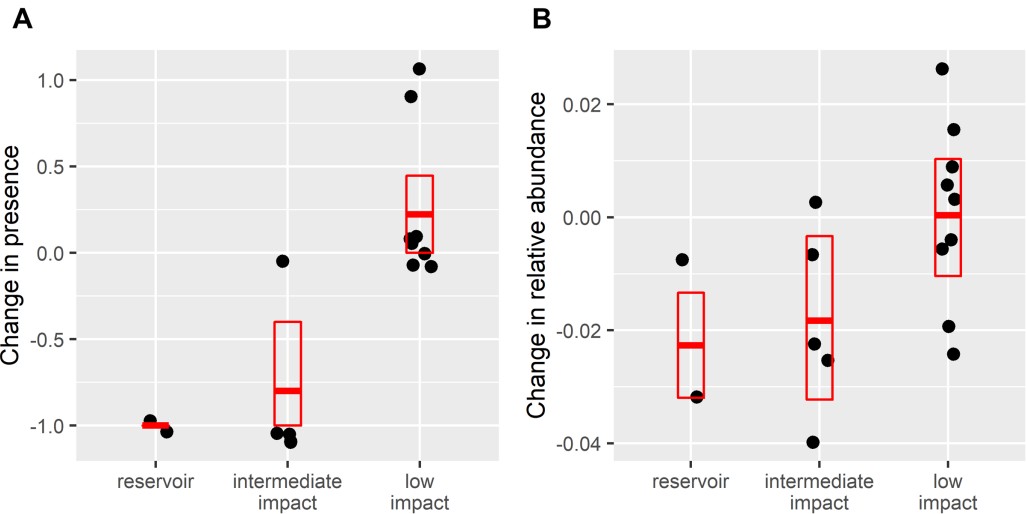

**Figure 3 Change in (A) giant otter presence and (B) relative abundance after dam construction.** Points are difference in values from 16 subzones where giant otters were sighted between 2011–2015 (before dam construction) and 2017–2019 (after dam construction). Boxplots show means and 95% confidence limits estimated *via* nonparametric bootstrap.

**Table 2 Generalized Additive Models used to explain patterns in giant otter from 2011–2019 in the eastern Brazilian Amazon.** Models ranked according to an Information criteria approach using AICc (Akaike Information Criterion corrected for small sample sizes). K, number of parameters; Δ, difference in AICc between the model with the lowest score and subsequent models; wi, the model selection weights. Models ordered by AICc values.

| Model | Model construction | K | $R^2_{adj}$ | Dev. Exp. (%) | AICc | | |
|---|---|---|---|---|---|---|---|
| | | | | | AICc | Δ | *Wi* |
| Presence-absence | | | | | | | |
| Full | Y~ BA: Zone + Dist. to house + $s$(Effort)+$s$(Subzone[†]) | 8 | 0.40 | 39.0 | 78.7 | 0.0 | 0.67 |
| Effort | Y~ $s$(Effort)+$s$(Subzone[†]) | 4 | 0.29 | 26.1 | 81.1 | 2.4 | 0.20 |
| Before-After Zone interaction | Y~ BA : Zone +$s$(Subzone[†]) | 6 | 0.30 | 30.9 | 83.0 | 4.4 | 0.08 |
| Zone | Y~ Zone + $s$(Subzone[†]) | 3 | 0.28 | 23.3 | 83.4 | 4.8 | 0.06 |
| Distance to house | Y~ Dist. to house +$s$(Subzone[†]) | 4 | 0.17 | 17.8 | 90.3 | 11.6 | 0.00 |
| Null | Y ~ intercept | 0 | 0.00 | 0.0 | 99.8 | 21.1 | 0.00 |
| Before-After | Y~ BA + $s$(Subzone[†]) | 2 | 0.02 | 3.7 | 100.3 | 21.6 | 0.00 |
| Relative abundance | | | | | | | |
| Full | Y~ BA: Zone + Dist. to house + $s$(Effort)+$s$(Subzone[†]) | 9 | 0.14 | 31.7 | −60.1 | 0.0 | 0.76 |
| Before-After Zone interaction | Y~ BA : Zone +$s$(Subzone[†]) | 8 | 0.08 | 18.7 | −56.5 | 3.6 | 0.13 |
| Distance to house | Y~ Dist. to house +$s$(Subzone[†]) | 5 | 0.09 | 13.2 | −54.7 | 5.4 | 0.05 |
| Zone | Y~ Zone + $s$(Subzone[†]) | 5 | 0.08 | 11.7 | −53.8 | 6.3 | 0.03 |
| Effort | Y~ $s$(Effort)+$s$(Subzone[†]) | 5 | 0.06 | 9.7 | −51.9 | 8.2 | 0.01 |
| Null | Y ~ intercept | 3 | 0.00 | 0.0 | −50.9 | 9.2 | 0.00 |
| Before-After | Y~ BA + $s$(Subzone[†]) | 4 | 0.02 | 6.1 | −50.9 | 9.2 | 0.00 |

**Note:**
$s$, Non-parametric smooth terms; †, Random effects; $R^2_{adj}$, Adjusted R squared for the model; Dev. Exp. (%), Percent of total deviance explained.
giant otters around hydropower dams, were conducted in regions that do not necessarily represent present day Amazonia *i.e.*, regions with no evidence of human occupation on the reservoirs banks (*Calaça & de Melo, 2017*). These conditions, with the presence of giant otters before dam construction and a low number of human communities were also highlighted as key conditions to maintain giant otters in dam reservoirs (*Rosas, Mattos & Cabral, 2007*). Our results corroborate the negative effects of human presence for this species as both most plausible models (for presence and relative abundance) of giant otters retained variables related to human disturbance (zone and distance to the nearest house).

Surprisingly, in a region with the lowest deforestation levels across the Brazilian Amazon (*INPE, 2021*), with low human density and large protected area coverage (*Kauano, Silva & Michalski, 2017*), we found that giant otters were still experiencing declines. The Amapa National Forest, one of the sustainable-use reserves surrounding our study region was considered to have a highly effective protection performance compared with a large spectrum of protected areas within the Brazilian Amazon (*Barber et al., 2012*). Yet, we show that the presence of protected areas *per se* is not sufficient to ameliorate direct and indirect negative effects of hydropower developments.

As giant otters are social animals, living in groups with up to 16 individuals with complex intra-social interactions (*Davenport, 2010*; *Duplaix, 1980*) and with a diet mainly based on fish that generates negative interactions with fisherman (*Michalski et al., 2012*; *Rosas-Ribeiro, Rosas & Zuanon, 2012*), more specific mitigation strategies to reduce negative effects caused by new hydropower developments must be developed. Additionally, dams affect the environment flooding terrestrial areas permanently, which can affect habitat use for giant otters as they build dens and maintain campsites along waterways (*Carter & Rosas, 1997*; *Duplaix, 1980*). During the initial phase of a new dam construction the removal of the riparian vegetation and the noise and movement of people can also reduce giant otter habitat use (*Calaça, Faedo & de Melo, 2015*). The increase in anoxia levels created by dams tend to increase mercury levels in the food web as well as change fish productivity and community (*Duplaix, Evangelista & Rosas, 2015*), all of which can negatively affect piscivorous species such as giant otters.

In our study changes to giant otter distribution and relative abundance after dam construction were caused by anthropogenic impacts at two scales—zones and subzones (houses). In the reservoir and intermediate anthropogenic impact zone both presence and relative abundance of the species reduced compared with the period before dam construction. The displacement of human riverside populations (*Fearnside, 2014*) that may move to previously uninhabited areas post dam construction could negatively affect giant otters (*Duplaix, Evangelista & Rosas, 2015*; *Rosas, Mattos & Cabral, 2007*). In our study, there was an alarming increase in houses in the intermediate impact zone, where the number of houses more than doubled after dam construction (from 0.14 to 0.35 houses per km). Thus, it is not surprising that giant otters reduced significantly in the intermediate impact zone. Although the number of houses reduced in the reservoir zone after the dam (from 1.78 to 1.51 houses per km), this reduction was clearly not sufficient to allow giant otters to reestablish the use of the reservoir zone, as we could not find the presence of the species even after 3 years of monitoring. As riverside people usually fish around

500 m from their houses in our study region (*Norris & Michalski, 2013*) it is likely that the density of houses (above 0.14 per river km) coupled with boat/canoe movement is an impediment for giant otters to occupy the two zones with higher human disturbance.

Our study used a unique dataset on the effects of hydropower dams on giant otters, demonstrating that even in areas with large protected area coverage and relatively low human density dam developments and associated social effects including human displacement may limit coexistence for a large apex-predator such as giant otters. We therefore recommend that the creation and maintenance of areas with low levels of anthropogenic disturbance to enable sustainable coexistence between energy demands and biodiversity across the Amazon should be mandatory in Environmental Impact Assessments. With the growing development of new and planned hydropower dams across the Amazon (*Anderson et al., 2019*; *Finer & Jenkins, 2012*), there is need to act rapidly and efficiently in order to avoid no return effects from these new developments in such a bio diverse ecosystem.

The implementation of minimization and mitigation actions to ameliorate negative impacts caused by dams remains limited for Amazonian vertebrates due to a lack of evidence as to efficacy of available options (*Bárcenas-García et al., 2022b*; *Berthinussen, Smith & Sutherland, 2021*; *Santos, Michalski & Norris, 2021*). For example, the installation of dam bypass channels could allow aquatic mammals to swim around dams and access habitats on both sides, which could be particularly relevant when multiple dams are located along the same river as in our study area. As giant otters are semi-aquatic animals, they can move on land and dam structures may not act as genetic barriers (*Duplaix, Evangelista & Rosas, 2015*). Yet, to date no studies evaluated the effects of installing dams bypass channels on freshwater mammal populations (*Berthinussen, Smith & Sutherland, 2021*). Perhaps one of the most important changes in dam reservoirs for this apex predator is the increase in anoxia levels, which increases mercury levels in the food web as well as changing fish productivity and assemblages (*Duplaix, Evangelista & Rosas, 2015*), all of which can affect piscivorous species such as giant otters. We are unaware of any studies that addressed such impacts on giant otters. For this long lived social species, additional population level studies on group size, social behavior and feeding habits must be developed in areas with new dam constructions both before and after environmental changes caused by dams.

## CONCLUSIONS

Our long-term study revealed that giant otters do not remain nor return to a reservoir with anthropogenic disturbance. Our study area, in one of the regions with the lowest deforestation levels across the Brazilian Amazon (*INPE, 2021*), with low human density and large protected area coverage (*Kauano, Silva & Michalski, 2017*), demonstrated that giant otters were still experiencing declines. Indeed, our results showed that the presence of protected areas *per se* is not sufficient to ameliorate direct and indirect negative effects of hydropower developments. Giant otters have a diet mainly based on fish that generates negative interactions with fisherman (*Michalski et al., 2012*; *Rosas-Ribeiro, Rosas & Zuanon, 2012*), which reinforces that more specific mitigation strategies to reduce

negative effects caused by new hydropower developments must be developed. Therefore, the creation and maintenance of areas with low levels of anthropogenic disturbance to enable sustainable coexistence between energy demands and biodiversity across the Amazon should be mandatory in future Environmental Impact Assessments.

## ACKNOWLEDGEMENTS

The Instituto Chico Mendes de Conservação da Biodiversidade (ICMBio) and the Federal University of Amapá (UNIFAP) provided logistical support. We are grateful to Alvino Pantoja Leal, Cremilson and Cledinaldo Alves Marques, Edinaldo and Davi Sousa, and Gilberto Souza for their invaluable assistance during fieldwork. We also thank all students, volunteer interns, and field assistants who participated on field activities over the study period. Fernando C. W. Rosas, Caroline Leuchtenberger, and Maria João Veloso Pereira provided suggestions on an earlier version of this manuscript. We also thank Mahendra Tomar, Lisa Davenport and two anonymous reviewers whose helpful comments improved this manuscript.

### Funding

This study was supported by an MSc scholarship from the Program PAEC OEA-GCUB and the Federal Agency for Support and Evaluation of Graduate Education, Brazilian Ministry of Education (Coordenação de Aperfeiçoamento de Pessoal de Nível Superior - CAPES process 88882.345624/2019-01) and by Idea Wild to Deborah Raffo. This research was funded by the National Academy of Sciences and the United States Agency for International Development through the Partnership for Enhanced Engagement in Research (award number AID-OAA-A11-00012) to Darren Norris and Fernanda Michalski. This study was also supported by the National Council for Scientific and Technological Development - CNPq (Process 305549/2018-9) to Sandra Hartz. This research was also supported by Conservation International – Brazil and the Walmart Institute – Brazil through the project "Support to the implementation of the Amapá National Forest", The Rufford Small Grants for Nature Conservation, the CNPq (Processes 477629/2011-3, 301562/2015-6, 403679/2016-8, 302806/2018-0), and Conservation, Food & Health Foundation to Fernanda Michalski. Fernanda Michalski receives a productivity scholarship from the CNPq (Process 310573/2021-1). The funders had no role in study design, data collection and analysis, decision to publish, or preparation of the manuscript.

### Grant Disclosures

The following grant information was disclosed by the authors:
Program PAEC OEA-GCUB.
Federal Agency for Support and Evaluation of Graduate Education.
Brazilian Ministry of Education: 88882.345624/2019-01.
Idea Wild.
National Academy of Sciences.

United States Agency for International Development: AID-OAA-A11-00012.
National Council for Scientific and Technological Development: 305549/2018-9.
Conservation International, Brazil.
Walmart Institute, Brazil.
Rufford Small Grants for Nature Conservation.
CNPq: 477629/2011-3, 301562/2015-6, 403679/2016-8, 302806/2018-0.
Conservation, Food & Health Foundation.
CNPq: 310573/2021-1.

## Competing Interests

Fernanda Michalski is a research associate from the Pro-Carnivores Institute. Fernanda Michalski and Darren Norris are Academic Editors for PeerJ. The authors declare that they have no competing interests.

## Author Contributions

- Deborah C. Dávila Raffo performed the experiments, analyzed the data, authored or reviewed drafts of the article, and approved the final draft.
- Darren Norris conceived and designed the experiments, performed the experiments, analyzed the data, prepared figures and/or tables, authored or reviewed drafts of the article, and approved the final draft.
- Sandra Maria Hartz conceived and designed the experiments, authored or reviewed drafts of the article, and approved the final draft.
- Fernanda Michalski conceived and designed the experiments, performed the experiments, analyzed the data, prepared figures and/or tables, authored or reviewed drafts of the article, and approved the final draft.

## Field Study Permissions

The following information was supplied relating to field study approvals (*i.e.*, approving body and any reference numbers):

Permissions to conduct research were approved by IBAMA and ICMBio (IBAMA/SISBIO permits 26653, 49632, and 69342).

## Data Availability

The raw data used in all analyses is available in the Supplemental File.

## Supplemental Information

Supplemental information for this article can be found online at http://dx.doi.org/10.7717/peerj.14287#supplemental-information.

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
