# Peer review of "Anthropogenic influences on the distribution of a threatened apex-predator around sustainable-use reserves following hydropower dam installation"

_PeerJ, doi:10.7717/peerj.14287_

## Round 0.1 · original submission · Major Revisions

Dear Dr. Michalski,

As per the comments from our expert reviewers, the study carried out is good and can be considered for publication in PeerJ. However, the manuscript has some lacunas which should be filled in a revision. Two reviewers have asked for minor revisions but one of them has suggested the major revision which seems to be justified. Please make all the mentioned corrections or justify your stand if does not match with the reviewer.

Revise and resubmit asap, Good luck.

·

Basic reporting

I believe the article meets the standards. I reviewed an earlier version which had some confusing elements, but those are now improved. I recommend publication and only have small corrections to English usage, which overall is excellent.

Experimental design

The BACI design is a nice application in this question, for the species, and new to my knowledge, as the authors claim.

Validity of the findings

I think the findings are new, credible in the analysis, and very interesting for conservation practitioners interested in the fate of this highly charismatic species.

Additional comments

I think this is an important study for the conservation of the species. Congratulations to all the authors on the novel design and work.

Minor English edits I recommend:

Abstract:

Line 28: “totaling” not totalizing

Intro:

Line 47, probably better style is to start with “To our knowledge, here we provide…”

Line 55: Add a comma after giant otters

Data analysis Section

Line 117: Standard English would have punctuation as follows:

“Two response variables were analyzed: (1) presence, and; (2) relative abundance….”

Discussion:

Line 166. Our “results” corroborate, (not “result”)

Reviewer 2 ·

Basic reporting

No comment

Experimental design

No comment

Validity of the findings

No comment

Additional comments

Brief not about Giant Otters: Conservation status and basic biological attributes like distribution, home range, habitat, feeding habits, and threats.
Line 42: Species which have had maximum impact on its range and population apart from otters
Line 57: Enlist conservation important species in the study area rather than just "vertebrates"
Line 74: "Sustainable reserves" please define. Also, mention anthropogenic activities permissible in these reserves.
Line 89-83: Historical distribution and population status of otters in the current study area.
Line 113: Mention the type of boat used for conducting the survey.
Line 191: What was the group size (Min-Max) of otters detected in the current study? Also was there any impact on the group size before and after the construction of the dam?
Line194: Elaborate on mitigation strategies according to the authors or comment on the limitation of the existing mitigation strategy on otters that have been deployed while construction of the dam.

Reviewer 3 ·

Basic reporting

The title suggests that the authors determined anthropogenic influences on otter distribution at dam sites. The manuscript clearly establishes the decline in giant-otter populations at dam sites in Brazil by using a BACI design by assessing populations before and after the construction of dams. The manuscript; however, provides limited information about the anthropogenic influences on otter habitat occupancy.

The language used in the paper is appropriate.

The introduction needs to include information on critical habitat variables that determine giant otter habitat selection and how dams by altering water-flow regimes and water levels affect the availability of suitable habitats for otters/aquatic mammals.

The discussion section needs to be reinforced with information on how these critical habitat variables are altered by the construction of dams and increased anthropogenic influences. The authors may like to refer to publication such as those of Calaca et al (2015) titled ‘Hydroelectric dams: the first responses from giant otters to a changing environment’; Carter & Rosas (1997) ‘Biology and conservation of the giant otter Pteronura brasiliensis.

The figures and tables included in the paper support the statements made in the publication. The authors can include an additional figure depicting on a map how critical variables determining otter habitat use changed before and after construction of dams.

The raw data supplied is also limited to presence of otters only.

Experimental design

The study addresses the objectives stated in the manuscript however it could be strengthened by including critical habitat variables determining habitat use by otters.

Conner et al (2016) ‘Evaluating impacts using a BACI design, ratios, and a Bayesian approach with a focus on restoration’ may be referred for giving greater clarity to the study.

Validity of the findings

The model developed by the authors clearly establishes the decline in otter populations at the reservoir and intermediate impact sites; however the same needs to be related to ecological variables that determine otter habitat preference and how the construction of dams effect these variables.

The raw data provided by the authors support the statements made by the authors; the same may be augmented with data on ecological variables that determine otter habitat preference.

Additional comments

The publication documents the decline of giant otter populations after construction of dam sites and may be considered for publication after incorporating suggested modifications.

---

## Round 0.2 · accepted · Accept

Dear Dr. Michalski,

It is my pleasure to inform you that as per the recommendation of our expert reviewers, the manuscript "Anthropogenic influences on the distribution of a threatened apex-predator around sustainable-use reserves following hydropower dam installation " - has been Accepted for publication in PeerJ.

This is an editorial acceptance and you will be contacted with the list of further tasks before publication. So, I request you to be available for a few days to make the necessary things asap.

Regards and good luck with your future submissions.

Reviewer 2 ·

Basic reporting

No comment

Experimental design

No comment

Validity of the findings

No comment

Additional comments

The authors have incorporated the suggested revisions.

Reviewer 3 ·

Basic reporting

No comment

Experimental design

The authors have made possible changes as is indicated in the manuscript and in their letter of rebuttal.

Validity of the findings

No comment

Additional comments

The authors have made changes as requested to the extent possible with existing information at their end.
The manuscript may be accepted for publication in your Journal as it meets the standards required.
I wish the authors all the best for their further endeavors.